# Green Synthesis and Antiproliferative Activity of Gold Nanoparticles of a Controlled Size and Shape Obtained Using Shock Wave Extracts from *Amphipterygium adstringens*

**DOI:** 10.3390/bioengineering10040437

**Published:** 2023-03-30

**Authors:** Daniela Torres-Ortiz, Guadalupe García-Alcocer, Achim M. Loske, Francisco Fernández, Edgardo Becerra-Becerra, Rodrigo Esparza, Marlen Alexis Gonzalez-Reyna, Miriam Estevez

**Affiliations:** 1Facultad de Química, Universidad Autónoma de Querétaro, Cerro de las Campanas s/n, Santiago de Querétaro 76010, Querétaro, Mexico; 2Centro de Física Aplicada y Tecnología Avanzada, Universidad Nacional Autónoma de México, Boulevard Juriquilla 3001, Santiago de Querétaro 76230, Querétaro, Mexico

**Keywords:** gold nanoparticles, green synthesis, nanomedicine, shock wave-assisted extraction, leukemia

## Abstract

In this study, green chemistry was used as a tool to obtain gold nanoparticles using *Amphipterygium adstringens* extracts as a synthesis medium. Green ethanolic and aqueous extracts were obtained using ultrasound and shock wave-assisted extraction. Gold nanoparticles with sizes ranging between 100 and 150 nm were obtained with ultrasound aqueous extract. Interestingly, homogeneous quasi-spherical gold nanoparticles with sizes between 50 and 100 nm were achieved with shock wave aqueous-ethanolic extracts. Furthermore, 10 nm gold nanoparticles were obtained by the traditional methanolic macerate extraction method. The physicochemical characteristics, morphology, size, stability, and Z potential of the nanoparticles were determined using microscopic and spectroscopic techniques. The viability assay in leukemia cells (Jurkat) was performed using two different sets of gold nanoparticles, with final IC_50_ values of 87 µM and 94.7 µM, reaching a maximum cell viability decrease of 80% The results do not indicate a significant difference between the cytotoxic effects produced by the gold nanoparticles synthesized in this study and vincristine on normal lymphoblasts (CRL-1991).

## 1. Introduction

For centuries, medicinal plant extracts have been widely used in traditional medicine to treat a diverse range of ailments. Currently, medicinal chemistry studies the properties and composition of extracts obtained from medicinal plants with the goal of determining the components present and their biological activity. In Mexican traditional medicine, the aqueous extract obtained from the bark of the Cuachalalate (*Amphipterygium adstringens*) tree is used to treat gastric ulcers, renal and hepatic pain, and constipation, and to reduce cholesterol levels [1]. Phytochemical studies of the extract have reported the presence of anacardic acid as the majority compound, particularly 6-pentadecylsalicylic acid (6SA). Both the methanolic extract and purified 6SA from *A. adstringens* have excerpted antiproliferative activity against breast and leukemia cancer cell lines [2,3].

There is also a constant search for novel drug development to treat cancer using nanotechnology. Specifically, novel metal nanoparticles (gold and silver) have unique spectroscopic, electric, optical, and magnetic properties that make them eligible as an alternative to be used as drug carriers. Additionally, gold nanoparticles (AuNPs) have been demonstrated to be safe because they have low toxicity compared to other metal nanoparticles. However, nanoparticle synthesis uses toxic stabilizing and reduction reagents that can make them incompatible with life, and the byproducts of synthesis pollute the environment [4,5]. 

Green chemistry focuses on the use and development of cleaner solvents, extraction, and synthesis methods [6,7]. Nanoparticle green synthesis can be achieved using plant extracts as a synthesis medium due to their reduced capacity, which depends primarily on secondary metabolite composition [8,9,10].

Plant extracts are widely known to be rich sources of a wide range of secondary metabolites. Therefore, every step of the extraction process can determine the obtention and preservation of these compounds. Extraction methods such as maceration are successful in retrieving secondary metabolites. Nonetheless, the extraction periods are long, and the extraction typically uses toxic organic solvents. On the other hand, heat extraction methods are effective in most cases but tend to degrade thermolabile compounds [4]. However, green chemistry is continuously developing novel low-key energy extraction methods using green solvents that are non-pollutive, biodegradable, and reusable, such as water and ethanol. Cavitation-induced extraction methods are widely used to disrupt the membrane and obtain secondary metabolites from plants. Examples are ultrasound and, more recently, shock wave-assisted extraction [11,12]. 

Shock waves generated in fluids (typically water) with a Mach number near one have been used in medicine and biotechnology for many years (see Appendix A) [13,14,15,16,17,18,19,20,21,22]. 

Well-known effects of the interaction of weak shock waves in fluids with matter are acoustic cavitation, shear stress, and spalling (i.e., the Hopkinson effect). Spalling is unlikely to appear when fragments are less than approximately 3 mm in size.

Cavitation is produced because the suspension contains microbubbles and nucleation sites. The positive pressure pulse of the shock wave compresses previously existing microbubbles, followed by fast bubble growth caused by the high pressure inside the bubble and outside pressure drop (Appendix A). Typically, bubbles collapse asymmetrically, and one side moves inward faster than the opposite side, forming a high-speed (up to 700 m/s) microjet of fluid that burrows through the bubble toward the posterior side of it [23,24,25,26,27,28,29,30,31,32].

If a shock wave passes through a suspension containing small particles, the bubble collapse generates microjets and fluid shear stress, crushing them and increasing the contact area exposed to the surrounding fluid. Additionally, microjets can lyse individual cells, producing the desired phytochemicals. 

So far, there is little information on the details of the phenomena involved in the shock wave-assisted extraction of compounds from plants [12]. Spatial pressure variations and acoustic cavitation produced by each shock wave are believed to be the main phenomena contributing to obtaining plant extracts. Furthermore, particles suspended in fluids enhance cavitation. Microbubbles appear and grow because the fluid–particle interfaces do not withstand the rarefaction produced by the negative phase of the shock waves. Surface roughness and hydrophobicity additionally reduce the adhesion between particle surfaces and the surrounding fluid, promoting bubble growth. Viscoelastic shear forces produced by high-speed microjets may also contribute to plant extraction. Tandem shock waves [20] (see Appendix A) were used in this study to enhance extraction efficiency. 

Ultrasound-assisted extraction (UAE) enhances the yield of extraction of secondary metabolites due to different mechanisms such as fragmentation, erosion, sonocapillary sonoporation, local shear stress, and dextrudation that occur when the ultrasound wave is applied [33]. Furthermore, parameters such as frequency and time of extraction can affect the integrity of the compounds in the extract [34].

The use of plant extracts to obtain metallic nanoparticles has two purposes. The first is the reduction in the metallic salt to its metallic pure state so that nanoparticles (NPs) can be formed. The second relies on the functionalization of the NPs themselves. As the reduction is occurring, the secondary metabolites from the extract become part of the NP structure, as is the case for AuNPs obtained using tartrate or citrate as a reducing agent [35]. The antioxidant capacity, as well as the total phenolic and flavonoid content of the extracts, are important because these factors lead to NP synthesis. 

The physicochemical characteristics of AuNPs, such as Z potential, NP size distribution, morphology, and size, are important to consider when performing a biological assay. For example, non-spherical 60 nm AuNPs produced hepatotoxicity in cell models, while spherical 60 nm AuNPs did not produce any hepatotoxicity; in contrast, spherical 20 nm AuNPs can cause liver damage in mice, while spherical 53 nm AuNPs administered to mice for 90 days did not produce marked toxicity [4,36,37]. Additionally, the use of plant extracts that have demonstrated a wide range of biological effects, such as antiproliferative activity in cancer cell lines, can functionalize AuNPs. However, a downside of the previously mentioned extracts is the toxic solvents with which they are obtained. 

The goal of this study was to compare *A. adstringens* extracts obtained by green extraction using green solvents (water and ethanol) along with green extraction methods (ultrasound and shock waves), and to study how different extraction methods can affect the green synthesis of AuNPs. The extracts were used for the green synthesis and functionalization of AuNPs, which were then characterized. We used physicochemical characteristics (UV–Vis, DLS, ZP, IR, DRX, and SEM) to remove those AuNPs that did not meet the size, morphology, and stability criteria. Additionally, the cytotoxic activity of AuNPs and *A. adstringens* extracts was determined in a Jurkat leukemia cell line and CRL-1991 normal lymphoblast cells. 

## 2. Materials and Methods

### 2.1. Materials 

A sample of *A. adstringens* (Schltdl) bark used for this study was obtained from a local market in the state of Querétaro, México, and identified as IBUNAM:MEXU:24829. Methanol (anhydrous 99.8%), ethanol (ACS reagent 99.5%), choline chloride (99%), betaine hydrochloride (98%), glycerol (ACS reagent, 99.5%), sucrose (99.5%), gold III chloride trihydrate (99.9%), Folin and Ciocalteu phenol reagent (2 M), gallic acid monohydrate (acs reagent, >98%), sodium carbonate (anhydrous, >99.5%), sodium hydroxide (>97%, pellets), quercetin (>95%), 2,2.diphenyl-1-picrylhydracyl (DPPH), neocuproine (>98%), ammonium acetate (>98%), (±)-6-hydroxy-2,5,7,8-tetramethylcromane-2-caboxylic acid (97%), and 2,2-azino-bis(3-ethylbenzothiazoline-6-sulfonic acid) were supplied by Merck KGaA, Darmstadt, Germany.

### 2.2. Obtaining Green Extracts

The *A. adstringens* bark was washed, dried, ground, passed through a sieve, and then stored at room temperature. In each extract, the amount of ground bark used was 67 mg/mL. Methanol and different mixtures of green solvents (water, water:ethanol 1:1, and ethanol) were used for each green extraction method. Macerate extracts were kept in a dark and dry place for 7 days. Ultrasonic-assisted extraction was performed in 10 mL beakers with the amounts of sample and solvent previously mentioned using a Cole-Palmer ultrasonic EW-08895-55 sonotrode (Vernon Hills, USA) at a frequency of 40 kHz and a power setting of 80 W for 15 s [38]. 

Shock wave extraction was performed using a Piezolith 2501-based experimental tandem shock wave generator (Richard Wolf GmbH, Knittlingen, Germany). The equipment produces shock waves via high-voltage (2–6 kV) discharges applied to 3000 piezoelectric crystals mounted on a hemispherical (radius = 345 mm) concave aluminum backing (Appendix A). The water-sealed layer of piezoelectric elements is located at the bottom of a Lucite water tub (base 675 mm × 675 mm, height 450 mm) with an XYZ positioner mounted on its top. The shock wave rate and voltage were fixed at 0.5 Hz and 5 kV, respectively. At this voltage, the peak positive pressure, measured with a polyvinylidene difluoride needle hydrophone (Imotec GmbH, Würselen, Germany) had an amplitude of 66.48 ± 1.49 MPa. All samples were exposed to 4000 shock waves (2000 events) in the tandem mode, using two different protocols. The water level and temperature were maintained at 40 mm above F and 20 °C, respectively. 

For shock wave exposure, samples containing *A. adstringens* and solvent were transferred into 4 mL cigar-shaped (length: 41 mm, diameter: 13 mm) polyethylene pipettes (Thermo Fisher Scientific, Waltham, MA, USA). Each vial was filled completely, heat-sealed, and positioned vertically along the axis of symmetry of the shock wave generator so that the center of curvature of the bottom of the transfer pipette coincided with *F* (Appendix A). The error in positioning the sample was estimated to be 0.5 mm. The shock wave-assisted extraction is achieved by the disruption of the herbal material as a consequence of acoustic cavitation.

After shock wave application, all samples were filtered, dried by lyophilization, and stored at −4 °C for further analysis.

### 2.3. Optimization of Parameters for Green Extraction Methods

Considering that both ultrasound and shock wave methods depend on cavitation, conditions should be adjusted to enhance this effect (see Appendix A) [39]. 

To optimize the extraction assisted by ultrasound, two variables were studied: extraction time and power. The extraction times tested ranged from 5 s to 90 s, increasing by 5 s per extract. For each extraction, the power of the ultrasound sonotrode was set at 40, 60, and 80 W. A preliminary antioxidant capacity (Appendix A) of the extracts was measured to set the parameters for the extraction (15 s and 80 W). 

Regarding shock wave-assisted extraction, the delays between the first and second shock waves were determined based on high-speed photography. Two criteria were followed. In the first case, the delay was fixed according to the time when most of the visible bubbles in the suspension started to collapse (i.e., when the visible density of the bubble cloud inside the transfer pipette began to decrease). 

Figure 1 shows a sequence of images obtained at 30,000 frames per second with a high-speed Motion Pro x4 (Integrated Design Tools, Inc., CA, USA) camera. Each image shows an ethanol-filled vial inside the water tank of the shock wave generator at different times during shock wave passage. The time *T* between one image and the next is 33.3 µs. At t = −*T*, the shock wave moving in the direction of the arrow had not reached the vial. The first visible bubbles appear at t = 0, indicating that the shock wave has passed through the water. The density of the bubble cloud inside the vial increased between t = 0 and t = 5*T*. During this period, most visible bubbles grew. The density decreases at t = 6*T*; thus, approximately 166 µs after the shock wave passage, in which the most visible bubbles have started to collapse. This time was chosen as the delay for one part of the experiments reported in this study. 

The diameter of a microjet emitted during shock wave-induced bubble collapse is known to be approximately one-tenth of the bubble diameter [19,24,25,32]. Even if there is no control over the size of the bubbles inside the vial, tandem shock waves generated at a specific delay selectively increase the collapse energy of bubbles having a certain diameter.

Assuming that larger microjets can cause more damage to large particles at the beginning of the treatment and that microjets with a smaller diameter may enhance cell lysis at the final stage of the treatment, a tandem shock wave application with three different delays for the same sample was also tested. The collapse energy of larger bubbles was boosted using a delay of 216 µs (~6.5 *T*). Analogously, a short delay of 116 µs (~3.5 *T*) was supposed to improve the pulverization of the particles at the final stage of shock wave application. The 166 µs delay (~5 *T*) was used during the intermediate phase of treatment.

Following this criterion, in the tandem mode with variable delay (SWVD), 200 events were administered at a delay of 216 µs, followed by 700 events at a delay of 166 µs, and finally 1100 events at a delay of 116 µs. 

The number of events (200, 700, and 1100) was chosen based on previous experiments documenting the particle size vs. the number of applied shock waves. A vial containing the *A. adstringens* suspension was exposed to up to 2000 shock waves at the above-mentioned pressure. Two drops were removed from the vial every 400 shock waves and analyzed under an optical microscope (Olympus Co., Tokyo, Japan). As an example, Figure 2 shows the particles before treatment and after receiving 2000 shock waves. Due to the large number of variables involved, this grouping should be considered empirical. Different combinations of delays and the number of applied shock waves could have been more efficient. Reporting the use of tandem shock waves with three different delays for each sample is justified because, in some cases, the results were better than those produced by tandem shock waves generated with the fixed 166 µs delay. Table 1, present the acronyms of the extract samples.

### 2.4. Antioxidant Capacity

The antioxidant capacity of all extracts was measured to determine which extract could be used as a reductive medium for the gold salt and to synthesize the AuNPs. DPPH was used to determine the free radical scavenging capacity of the extracts. Briefly, a 1 µL sample was added to 2.9 mL of the DPPH solution and incubated for 60 min in the dark. The absorbance was registered at 515 nm, and the results were expressed in Trolox mEq [40].

The CUPRAC assay was used to determine the antioxidant capacity by electrochemistry [41]. A CuCl_2_ solution with a concentration of 3 × 10^−3^ M in distilled water was prepared. Additionally, a solution of neocuproine at 6 × 10^−3^ M in ethanol was prepared. To control the pH of the primary solution, a 1.2 M ammonium acetate buffer solution was prepared (pH = 7), and the pH was adjusted by adding 1.2 M HCl and 1.2 M NaOH as required. To obtain the standard curve, a Trolox solution was prepared at different concentrations (1 × 10^−4^, 2 × 10^−4^, 4 × 10^−4^, 6 × 10^−4,^ and 8 × 10^−4^ M) in ethanol to obtain the antioxidant capacity as a function of Trolox equivalents.

For the standard curve, 2 mL of each solution was prepared and added as follows: a solution of CuCl_2_, a solution of neocuproine, a solution of ammonium acetate buffer, and a solution of Trolox and distilled water to achieve a final volume of 10 mL. The final solution was stirred for 15 min and bubbled with N_2_ for 10 min at room temperature. The same steps were followed to evaluate the antioxidant capacity of the extracts replacing the Trolox addition.

### 2.5. Total Phenol and Flavonoid Content

The total phenol and flavonoid contents were determined because they actively participate as reduction agents of gold tetrahydrochloride; therefore, AuNPs can be synthesized. Briefly, total phenol content (TPC) was determined using gallic acid as a standard, adding 150 µL of gallic acid or sample solution to 1 mL of the Folin and Ciocalteu reagent (1:10) and incubating for 5 min, followed by the addition of 800 µL of sodium bicarbonate and incubation for 60 min. Absorbance was registered at 764 nm, and the results were registered in gallic acid equivalents [42]. The total flavonoid content (TFC) was determined using the well-known Zhishen method. A 50µL sample was added to 30 µL of NaNO_2_ (7.5%) and incubated for 3 min. After this, 30 µL of AlCl_3_ was added and incubated for 15 min. Absorbance was recorded at 510 nm and the results were registered in catechin equivalents [43].

### 2.6. Green Synthesis of AuNPs

To obtain AuNPs, the extracts were used as a reduction medium. One mg of the extract was resuspended in 1 mL of distilled water, and a solution of 10 mM gold tetrahydrochloride was prepared. One mL of the extract solution was added to 1 mL of the gold tetrahydrochloride solution. The synthesis was performed for 24 h at room temperature [44]. The AuNPs acronyms of the samples, are listed on Table 1.

### 2.7. AuNP Characterization

A UV—Vis spectra of the extracts was registered. Then, to determine if the synthesis of AuNPs was performed, the reaction was monitored by UV–Vis with a 1600 PC VWR spectrophotometer (Delaware, PA, USA). The plasmon resonance absorbance at 540 nm determined the presence of AuNPs in the extract, while its absence indicated that the synthesis of AuNPs was not performed. Then, the AuNPs were centrifuged for 10 min with distilled water at 12,000 rpm and a cleanse was repeated by triplicate and resuspended in distilled water, and then they were stored afterward at −4 °C for further experiments [44]. The size distribution of the AuNPs was registered by dynamic light scattering (DLS, Litesizer 500, Anton Paar, Austria). One hundred µL of the AuNP solution was dispersed in 1 mL of distilled water. A phosphate buffer was used to adjust the pH to 7.4. The analysis was conducted at 25 °C in reusable cuvettes (n = 3) [45]. To determine the stability and dispersion of the AuNPs, Z potential was measured (DLS, Litesizer 500, Anton Paar, Austria). A total of 100 µL of AuNPs was dispersed in 1 mL of water at different pHs (4, 7.4, and 11) with n = 3 [45]. AuNPs’ morphology and size were determined by scanning electron microscopy (SEM) and scanning transmission electron microscopy (STEM) analysis using an SEM/STEM Hitachi microscope (Hitachi High-Tec Tokio, Japan). For SEM analysis, the samples were dispersed and laid on carbon-coated copper holders, and for STEM analysis, a drop of the sample’s solution was dropped over the carbon-coated copper grids. Equipment employed to acquire the X-ray photoelectron spectra was assembled by Intercovamex (Morelos, Mexico); the experimental data collection instrumentation consists of an XR5 monochromatic XR5 monochromatic Al Kα1 (hv = 1486.7 eV) X-ray source and an Alpha110 hemispheric spectrometer with a seven-channel electron multiplier detector provided by ThermoFisher (East Grinstead, UK). The test was performed at 9 × 10^−11^ Torr with a pass energy of 10 eV. The peak-fitting analysis of the photoemission spectra was accomplished using the AAnalyzer^®^ software. Additionally, to determine the AuNP concentration obtained per mL of extract, a thermogravimetric assay was conducted on a Mettler/Toledo TGA/DSC 2+ thermal analyzer (Mettler Toledo, Ohio, USA). The analysis was performed in an air atmosphere, with a heating rate of 10 °C/min between 30 and 700 °C. Moreover, the functional groups present in the extracts, as well as the AuNPs synthesized in the extract, were reported using a Perkin Elmer Spectrum Two FTIR (PerkinElmer, Massachusetts, USA) spectrometer at 400–4000 cm^−1^. Finally, X-ray (XRD) patterns were obtained with a Rigaku Ultima IV diffractometer (Tokyo, Japan) with Cu Kα radiation (40 kV, 15 mA, 1.54051 Å).

### 2.8. Viability Assay in Leukemia Cell Lines

#### 2.8.1. Cell Culture

Jurkat cells (ATCC No. TIB-152) were obtained from the American Type Culture Collection (ATCC, Manassas, VA, USA) and were grown at 37 °C in a humidified atmosphere of 95% air and 5% CO_2_, with RPMI 1640 medium (ATCC, No. 30-2001) containing 10% fetal bovine serum (FBS, ATCC, No. 30-2020) and 2% antibiotics (penicillin and streptomycin sulfate).

#### 2.8.2. Cytotoxicity Assay with Trypan Blue

To determine the number of viable cells present in a cell suspension, we used the dye exclusion test, which is based on the principle that live cells possess intact cell membranes that exclude certain dyes, such as trypan blue, while dead cells do not [46]. To perform the viability assay, 1 part of 0.4% trypan blue was mixed with 1 part of the cell suspension, and the mix was incubated for 3 min at room temperature. Next, 10 μL of the solution was transferred into a hemocytometer placed on the stage of a binocular microscope. The unstained (viable) and stained (nonviable) cells were counted. The total number of viable cells per mL of aliquot was obtained by multiplying the total number of viable cells by 2, while the total number of cells per mL of aliquot was obtained by adding up the total number of viable and nonviable cells and multiplying by 2. The percentage of viable cells was calculated as follows:(1)$$\%\;\mathrm{of}\;\mathrm{viable}\;\mathrm{cells}=\frac{\mathrm{total}\;\mathrm{number}\;\mathrm{of}\;\mathrm{viable}\;\mathrm{cells}\;\mathrm{per}\;\mathrm{mL}\;\mathrm{of}\;\mathrm{aliquot}}{\mathrm{total}\;\mathrm{number}\;\mathrm{of}\;\mathrm{cells}\;\mathrm{per}\;\mathrm{mL}\;\mathrm{of}\;\mathrm{aliquot}}\times 100$$

Jurkat cells were treated with the following concentrations of AuNPs in µg/mL: 0.98, 3.94, 7.88, 11.82, 15.76, and 29.55 (5, 20, 40, 60, 80, 150, and 300 μM) for 24 h. Additionally, Jurkat cells were treated with 10, 20, 100, 200, 300, 500, and 1000 μg/mL extracts. To validate the experiment, 1 μM of ethanol and H_2_O were used as negative controls and 10 μM of vincristine was used as a positive control, and 90 µM was the concentration to induce 100% of cell death. Once the relative inhibitory concentration reached a value of 50 (rIC_50_), the concentration that reduces 50% of cell viability of each AuNPs and extract was determined, and such concentrations were tested in CRL-1991 cells to determine if they had the same cytotoxic percentage.

#### 2.8.3. Statistics

Data were normalized to the maximum signal, where dose-response and concentration-response curves were fitted with a three-parameter logistic model by non-linear regression using Graphpad Prism 8 to determine the concentration that induces 50 cell deaths. To determine significant differences between groups, two-way ANOVA and Bonferroni’s multiple comparison tests were performed in Graphpad Prism 8. Significant differences are represented as different letters (*p*-value ≤ 0.05).

## 3. Results and Discussion

### 3.1. Antioxidant Capacity

The antioxidant capacity and the total phenolic and flavonoid contents were determined for all the extracts. These parameters are important for the green synthesis to be delivered, because the antioxidant compounds present in the extracts lead the reduction of Au^+3^ to Au^0^. The methanolic extracts obtained with the green extraction methods had the lowest antioxidant capacity and total phenolic and flavonoid content of all the extracts.

In bark extracts, TPC and the antioxidant capacity have been shown to be thermolabile when obtained by heat-dependent methods such as ultrasound baths, presenting a low yield of extraction [47]. Results show that the TPC yield of extraction was solvent-dependent. Table 2 shows that aqueous ultrasound-assisted extraction was 91% higher than Mac-MeOH, while ethanol:water ultrasound-assisted extraction was 12.9% lower than Mac-MeOH. Conversely, shock wave-assisted extraction was delay-dependent. The TPC in samples exposed to tandem shock waves generated with a constant delay were 76.1% higher than Mac-MeOH. This value decreased when the variable delay was used but was still 29.3% higher than Mac-MeOH. Regarding the antioxidant capacity, once again, aqueous ultrasound-assisted extraction had the highest yield of extraction.

The results from both DPPH and CUPRAC assays showed that the yield of green extraction was higher in antioxidant capacity compared to the yield obtained by the traditional extraction method Mac-MeOH [19,24,25,32,47].

### 3.2. Green Synthesis of AuNPs

To obtain AuNPs with sizes ranging between 50–150 nm, different extracts from *A. adstringens* were used to determine the one that could deliver AuNPs with the characteristics required. In all cases, the absorbance of the AuNP plasmon was registered by analyzing the samples with UV–Vis. In Figure 3, the UV–Vis spectra of the samples exhibit an absorption band between 500–600 nm that is characteristic of AuNPs. Surface plasmon resonance (SPR) is the result of the oscillating surface electrons that resonate with the incident wavelength. In Figure 3a extract spectra from four different samples was registered to ensure that there is no absorption band in the range of the SPR absorption. In Figure 3b, the spectra of AuNPs posterior to cleanse are registered. For the methanolic macerate and aqueous ultrasound-assisted extract, the spectra exhibited a broad band with maximum absorption at 580 nm, which confirms the presence of AuNPs. Additionally, in both samples, the intensity of the absorption band is weak, which can be attributed to a low AuNP concentration. Conversely, the spectra of the AuNPs synthesized using the shock wave-assisted extracts have a well-defined maximum absorption band at 550 nm, suggesting the presence of AuNPs [48]. This band also has a higher intensity, which is associated with a higher AuNP concentration compared to the rest of the samples [49,50]. Nevertheless, in all samples, there is a wavelength shift in the plasmon due to the different sizes of the AuNPs synthesized [49]. Importantly, in regard to NPs, the absorption band of the SPR reveals a displacement to smaller wavelengths when the NP size is smaller, and to bigger wavelengths when the particle size is bigger [51].

### 3.3. AuNP Characterization

SEM images were obtained to observe the size and morphology of the AuNPs in each sample. These two parameters are important when biological assays are involved because toxicity can depend on the shape and size of AuNPs. For example, AuNPs of different sizes and shapes inoculated into an in vivo model of Wistar rats demonstrated that sphere-shaped AuNPs of 15 nm are more toxic than 60 nm AuNPs and that stars and triangles of 60 nm are hepatotoxic [4]. Figure 4 shows the SEM images of 12 AuNPs that met the requirements to be analyzed by this technique. After analysis, we determined which samples were viable to continue with the characterization and biological assays. The SEM images were obtained by adding the secondary electrons (SE) plus backscattering electron signals (HA) to better appreciate the contribution of the organic matrix and the AuNPs, with the exception of the Au@Mac-MeOH (Figure 4a) sample, which was obtained with SE (grayscale). The green color in the SEM images corresponds to the AuNPs embedded in the extract’s organic matrix (red color). In the image, the Au@Mac-MeOH sample shows heterogeneous NPs smaller than 50 nm, primarily quasi-spheric. 

Most gold nanoparticles analyzed by SEM had no important use in this study (see Appendix A) because they were oversized, undersized, or the shape was inadequate for biological assays. Nevertheless, star- and triangle-shaped and small-sized (<30 nm) AuNPs can be used for catalytic means. 

Conversely, quasi-spherical gold nanoparticles with suitable sizes can be described as follows. Figure 4b Au@US-W contains the largest NPs of the three samples, ranging between 100–150 nm and being primarily faceted in shape, but also having triangles larger than 500 nm. Interestingly, Figure 4c Au@SWCD-EW and Figure 4d Au@SWVD-EW show a high density of AuNPs, mostly quasi-spherical and faceted AuNPs in Au@SWCD-EW, and a more heterogeneous distribution of quasi-spherical and triangular shapes in Au@SWVD-EW, with sizes of 50 nm and 50–100 nm, respectively. The size of the AuNP samples agrees with the results obtained from the antioxidant capacity. Because the US-W extract had the higher antioxidant capacity of the four samples, it can reduce more Au^+3^ to Au^0^, resulting in larger NPs. Furthermore, Mac-MeOH extract exhibited the lowest antioxidant capacity, thus synthesizing the smaller AuNPs, as seen in Figure 4a. Characteristics such as the shape and size of the AuNPs obtained by green synthesis depend on the secondary metabolites present in the extract used as a reduction medium. Concurrently, the constituents obtained in the extract vary with the extraction method and solvents used. Thus, in most cases, the shape and size of AuNPs are difficult to control. AuNPs obtained with different plant extracts using the same methodology as those in this study have achieved AuNPs with a size of 30–50 nm on average; however, in some cases, up to 200 nm AuNPs can be obtained [52,53,54].

The size distribution of the AuNPs is important to determine the control of the synthesis reaction, as well as to predetermine if the AuNPs obtained are viable for use in in vitro assays. 

Contrasting results of the AuNP size distribution and the polydispersity index (PDI) were obtained by DLS, which indicates the presence of aggregates in the solution. In general, all AuNP samples are polydisperse, as the PDI is >0.1%, with Au@MeOH having the greatest PDI = 28.57% and Au@SWCD-EW exerting the lowest PDI = 15.91% (Figure 5). These PDI results suggest the presence of AuNP aggregates, which can be confirmed by the contrasting results of the particle size distribution obtained by DLS and the results of the micrographs obtained by SEM [55]. In Figure 5, DLS histograms show an overall greater particle size distribution of the AuNPs compared to the ones registered by SEM in Figure 4. While DLS showed an average particle size of 125, 110, 75, and 85 nm for Au@Mac-MeOH, Au@SWCD-EW, Au@SWVD-EW, and Au@US-W, respectively, the SEM analysis showed an overall particle size range between 50 and 100 nm for Au@SWCD-EW, Au@SWVD-EW, and Au@US-W, and <50 nm for Au@Mac-MeOH. 

Figure 6 shows the bright-field scanning transmission electron microscopy (BF-STEM) images of the samples, mainly exhibiting mass-thickness contrast. The particles showed quasi-spherical morphologies with an average size of 53.64 and 42.90 nm (histogram) for Au@SWCD-EW and Au@US-W, respectively, corroborating the particle size measured by SEM. The formation of an organic coating from the extract over the AuNPs can also be observed clearly. This organic coating does not allow the NPs to agglomerate. Annular dark-field (ADF)-STEM images exhibit Z-contrast; consequently, only the contribution of AuNPs can be appreciated. The images show that the nanoparticles are dispersed and not agglomerated.

The stabilities of the AuNPs at different pH were determined via the Z potential. While NPs that have potentials between +30 and −30 mV are considered unstable and tend to form aggregates, NPs with potentials > +30 mV and <−30 mV can be considered stable colloids. In Figure 7, Au@MeOH at all pH values had a Z potential <−30 mV, indicating a stable colloid. Regarding the Au@SWCD-EW, Au@SWVD-EW, and Au@US-W samples at pH = 4, the Z potential was >−30 mV, indicating an unstable solution. Notably, Au@SWCD-EW and Au@US-W exhibited Z potential values <−30 mV when the solution pH was 7.4. 

Considering the evidence collected from the SEM, DLS, and Z potential analysis, the AuNPs that were suitable for biological analysis were Au@SWCD-EW and Au@US-W because they contain primarily quasi-spherical NPs ranging between 50 and 100 nm and exhibit a stable colloid suspension at pH = 7.4. 

Before using these AuNPs in biological assays, it was important to determine the AuNP quantity present in the sample for which the thermogravimetric analysis was conducted. In Figure 8, the results for both samples (Au@SWCD-EW and Au@US-W) show a minor percentage decrease between 80 and 100 °C, corresponding to the moisture of the sample. A high percentage decrease was registered between 110 and 125 °C, which can be attributed to organic compound decomposition due to high temperature, as has been previously reported with metallic nanoparticles [56,57]. The total weight loss percentage for Au@SWCD-EW was 99.48%. The remaining weight corresponded to 0.3110 mg, which was 7.8 µg/µL. For the Au@US-W sample, the total weight loss was 99.57% and the AuNP concentration was 6.1 µg/µL. 

In order to determine which functional groups were potentially involved in AuNP synthesis, we registered the FTIR spectra of AuNPs and extracts for the samples Au@SWCD-EW, SWCD-EW, Au@US-W, and US-W. In Figure 9a, the spectra of Au@SWCD-EW and SWCD-EW almost overlap. It is important to mention that the Au@SWCD-EW sample was deposited onto a glass coverslip, and bands in the 2900–2850 cm^−1^ range correspond to SiO vibrations of the coverslip on which the sample was deposited. Absorption bands in the spectra from Au@SWCD-EW and the SWCD-EW extract are similar in the region between 1600–1100 cm^−1^; nevertheless, these differences are due to the intensity of the bands. A broad band between 3700 and 3050 cm^−1^ (centered at 3300 cm^−1^) can be observed in both the extract and AuNP spectra due to -OH stretching vibrations, which can be related to secondary metabolites such as phenols, carboxylic acids, and polyol compounds, which are known to stabilize AuNPs [58]. Specifically, in the Au@SWCD-EW spectra, a sharp absorption band at 1710 cm^−1^ appears, which is characteristic of cyclic amides that are known to participate in the stabilization and synthesis of AuNPs [9]. The sharp medium absorption band at 1600 cm^−1^ corresponds to carboxylate ions present in both the extract and AuNPs. Absorption bands between 1400 and 1300 cm^−1^ are characteristic of aromatic ring stretching vibrations, categorically in the AuNP spectra. An increase in the intensity of the bands in this area indicates a change in the substitution pattern of aromatic compounds that are well-known to interact with AuNPs [59]. Bands arising from C-O stretching and OH bending of carboxylic acids appear near 1320–1100 cm^−1^ for both the extract and AuNP spectra. Broad medium absorption bands are registered between 1270 and 1230 cm^−1^ [59]. 

Conversely, regarding the ultrasound extract and AuNP spectra in Figure 9b, as in shock wave samples, absorption bands were observed between 3700 and 3050 cm^−1^ (centered at 3330 cm^−1^), suggesting the presence of polyol compounds as well as carboxylic acids. A sharp medium absorption band at 1630 cm^−1^ is shown in the extract and AuNPs, corresponding to carboxylate ions from carboxylic acid. Additionally, bands between 1300 and 1000 cm^−1^ appear due to bending OH and stretching C-O interactions. Additionally, two strong absorption bands are observed in 875 and 800 cm^−1^, which correspond to C-H out-of-plane bending of aromatic compounds that typically appear in the region between 900–650 cm^−1^. Importantly, unlike Au@SWCD-EW, the Au@US-W spectra does not show the 1700 cm^−1^ absorption band, which can be attributed to using different extraction methods to obtain the *A. adstringens* extract and can be supported by the concentration discrepancy of flavonoids and phenols in the extracts. Additionally, to confirm the band position, second derivatives of both extract and AuNP FTIR spectra were calculated (Figure 9c,d), in which band positions (Figure 9a,b) are more notorious.

The X-ray diffraction (XRD) patterns of Au@SWCD-EW and Au@US-W are shown in Figure 10, which clearly illustrates that the synthesized AuNPs are crystalline in nature. Both XRD patterns show diffraction peaks at 2θ values of 38.22°, 44.42°, 64.63°, and 77.64°, which can be indexed to the crystallographic planes (111), (200), (220), and (311), respectively, of the face-center cubic (fcc) lattice of Au (JCPDS 65-8601). There is a marked difference in the broadening of Bragg reflections due to the size of the crystallites and the presence of facetted particles in both samples (inset figure). Rietveld refinement profile of XRD patterns was computed for crystal structure analysis [60]. As can be observed, there is a good agreement between the experimental data and the calculated refinement. The average obtained of the crystallite sizes were 18.97 ± 2.47 nm and 39.91 ± 5.75 nm for Au@SWCD-EW and Au@US-W, respectively. These results indicate a smaller crystallite size in the sample synthesized using shock wave-assisted extraction and ethanol:water, as previously shown by SEM; therefore, most of the synthesized nanoparticles are polycrystalline. There is a Bragg reflection that does not correspond to the Au structure, which is located at 2θ values of 31.72° and can be indexed to the NaCl structure (JCPDS 75-0306). This compound is a waste product during the synthesis process and can be removed by successive clean washings. 

The surface composition of gold nanoparticles was examined by XPS. Appendix A shows the survey spectra of the samples with signals from Au 4f, Si 2p, C 1s, and O 1s. The high-resolution spectrum of the gold 4f core level, shown in Appendix A, was deconvoluted with two doublet peaks corresponding to the spin-orbit coupling (Au 4f 7/2 and Au 4f 5/2) and revealed partial oxidation of the gold surface. The first and most important doublet peak (BEs of 84 eV) is related to elemental gold (Au^0^), while the other is related to a stable gold oxidation state of Au^3+^ (BE of 87.3 eV). 

On the other hand, the presence of carbon and oxygen in each sample is attributed to the organic compounds surrounding the AuNPs.

### 3.4. Biological Assays

For the biological assays, a concentration of 60 µM of AuNPs was considered to be a medium concentration because it has produced 80% of the proliferation decrease in leukemia cells previously [61].

#### Viability Assay in Leukemia Cells Treated with AuNPs (Au@SWCD-EW and Au@US-W), SWCD-EW, and US-W Extracts

The cytotoxic and antiproliferative activity of several NPs has been widely reported over different types of cancer cells. According to the physicochemical characterization of AuNPs, there were only two samples (Au@US-W and Au@SWCD-EW) that met the size, shape, and stability requirements for gold nanoparticles to be tested in biological assays. To determine the effect of AuNPs on leukemia cell viability, different concentrations of AuNPs, as well as the extract, were tested (n = 3). Both extracts reduced the Jurkat viability in a dose-dependent manner. In Figure 11a, US-W extract rIC_50_ was obtained at 202 μg/mL, while the rIC_50_ SWCD-EW extract was 234 μg/mL. Both US-W extract and SWCD-EW extract at 1 mg/mL inhibited 80% and 75% of viability, respectively, after 24 h of treatment. 

In addition, it has been reported that macerated methanolic extract from *A. adstringens* can except the total growth inhibition of different cancer in vitro models such as ovarian adenocarcinoma (NCl/ADR resistant cells), kidney adenocarcinoma (786-O cell), and prostate cancer (PC-3 cells), at a concentration of 27.5 ± 0.7 µg/mL. Nevertheless, concentrations >250 µg/mL were required to achieve total growth inhibition in chronic myeloid leukemia cells (K562 cells) [62]. 

Interestingly, AuNPs obtained by green synthesis using ultrasound and shock wave extraction (Figure 11b) reached rIC_50_ at a lower concentration compared to that of the extracts. Both Au@US-W and Au@SWCD-EW decreased Jurkat viability in a dose-dependent manner, reaching rIC_50_ values of 17.14 µg/mL (87 μM) and 18.66 µg/mL (94.7 μM), respectively, after 24 h of treatment. The maximum inhibition of cell viability was 75% for both AuNPs. However, Alam–Escamilla reported a decrease in proliferation at an rIC_50_ of 150 µM in chronic myeloid leukemia treated with 6-pentadecyl salicylic acid (6-SA) purified from *A. adstringens* hexane extract after 48 h of treatment [63]. Additionally, purified 6-SA from *A. adstringens* has been administered to in vivo cancer models (4T-1 tumor). For example, the tumor volume in mice treated with 6 mg/kg was reduced by 52% after 3 weeks of treatment [64].

Although AuNPs have been reported to induce oxidative stress and, thus, phosphorylation of p48 to trigger apoptosis in many cancers [65], it is necessary to determine which type of cell death is induced by Au@US-W and Au@SWCD-EW. 

Both US-W and SWCD-EW extracts have nonenzymatic antioxidant capacity, and it is important to consider that antioxidants at high concentrations may cause direct damage to DNA and further cell death [66].

Both negative controls killed the same percentage of CRL-1991 cells as Jurkat cells, and both controls induced significantly lower cell deaths than all treatments. In addition, vincristine 10 μM induced cell death with no marked differences with respect to either AuNPs or extracts. Both extracts and AuNPs induced cell death, but 17.14 µg/mL (87 μM) Au@US-W achieved the maximum cytotoxic effect, showing a significant difference with respect to 234 μg/mL SWCD-EW only. As shown in Figure 12, no AuNPs or extracts induced 50% cell death, as observed in Jurkat cells, which indicates that at least lymphoblast normal cells are resistant to such treatments, and leukemic cells are more sensitive. Changes in the redox balance modify the proliferation and survival rate in cancer cells, while in normal cells, no significant change is observed [67], which may explain the relative resistance of CRL-1991 cells to cell death by extracts because they have antioxidant activity. However, AuNPs may promote reactive oxygen species generation to trigger cell death [68] in both cell lines, but the concrete mechanism of cell death was not determined in this work. Further studies of cell death mechanisms will help to understand why CRL-1991 cells are less sensitive to cell death. In a hypothetical chemotherapeutic treatment with both AuNPs and extracts, the cytotoxicity may be less aggressive in hematopoietic cells, but different cell lines should be included in future studies. AuNPs were well-characterized and are good candidates for more extensive biological studies regarding cancer therapy. 

## 4. Conclusions

According to our results, *A. adstringens* water-ethanol extracts obtained with shock wave-assisted extraction achieved homogeneous quasi-spheric shaped and size-controlled AuNPs (Au@SWCD-EW) ranging between 50 and 100 nm. In contrast, ultrasound-assisted extraction to obtain aqueous extracts achieved bigger AuNPs (Au@US-W) ranging between 100 and 150 nm, which had irregular and triangular shapes. It is important to outline that even though both shock wave and ultrasound-assisted extraction methods produce cavitation, the synthesis of homogeneous shapes and sizes of AuNPs is only achieved using shock wave extracts. This can be attributed to the ability of shock wave-assisted extraction to produce different chemical compounds, as demonstrated in the TPC and TFC quantitative assays.

Additionally, in vitro biological assays confirmed that overall, AuNPs (Au@SWCD-EW and Au@US-W) and extracts (SWCD-EW and Au@US-W) reduce the viability of leukemia cells. Most importantly, Au@US-W achieved the rIC_50_ at 17.14 µg/mL (87 μM) and Au@SWCD-EW at 18.66 µg/mL (94.7 μM). The latter had a lower cytotoxic effect in normal lymphoblasts. This may be attributed to the homogeneous quasi-spherical and size-controlled Au@SWCD-EW and, therefore, could be a safer model for developing cancer therapy. Future perspectives for this work are that the AuNPs obtained could be used as safe drug nano-carriers. Finally, to our knowledge, there are no reports to date on AuNPs synthesized from *A. adstringens* extracts obtained by shock wave- or ultrasound- (using sonotrode) assisted extraction and their effect on cellular viability and cytotoxic activity on leukemia cells. 

## Figures and Tables

**Figure 1 bioengineering-10-00437-f001:**
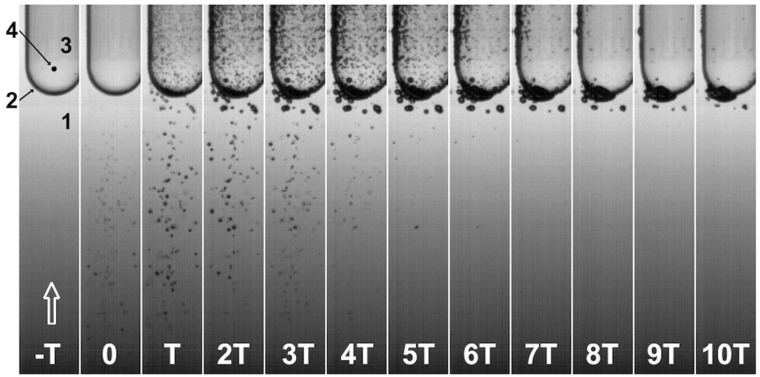
Sequence of high-speed images (*T* ≈ 33.3 µs) showing cavitation bubbles inside a sealed vial containing ethanol after exposure to a single-pulse shock wave with a positive peak amplitude of approximately 66 MPa. The arrow shows the direction of the shock wave, which propagated through water (1), penetrated the vial at its bottom (2), and continued its path through the ethanol (3). The focus F was located at the center of curvature of the vial (4). The first frame where bubbles could be observed was t = 0.

**Figure 2 bioengineering-10-00437-f002:**
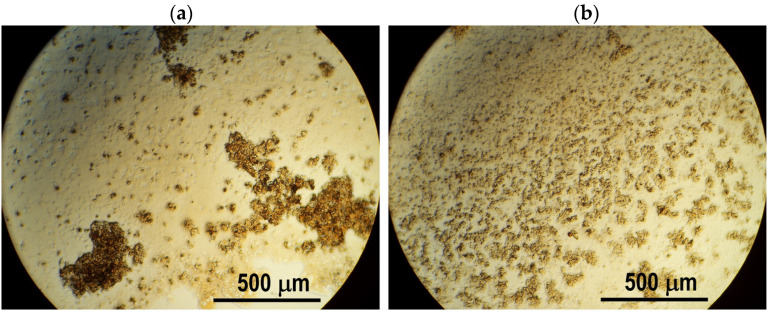
Micrograph of *A. adstringens* particles (**a**) before initiating shock wave application and (**b**) after receiving 2000 single-pulse shock waves with a positive pulse amplitude of approximately 66 MPa.

**Figure 3 bioengineering-10-00437-f003:**
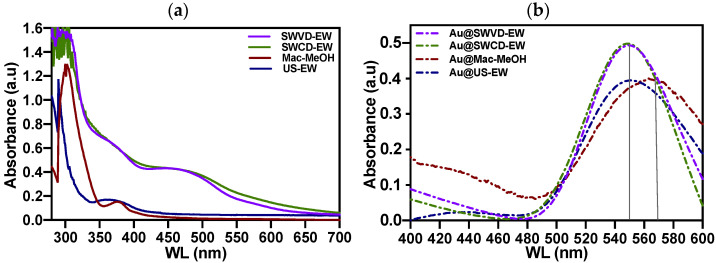
UV–Vis analysis of (**a**) *A. adstringens* extracts; (**b**) AuNPs synthesized using different extracts from *A. adstringens*.

**Figure 4 bioengineering-10-00437-f004:**
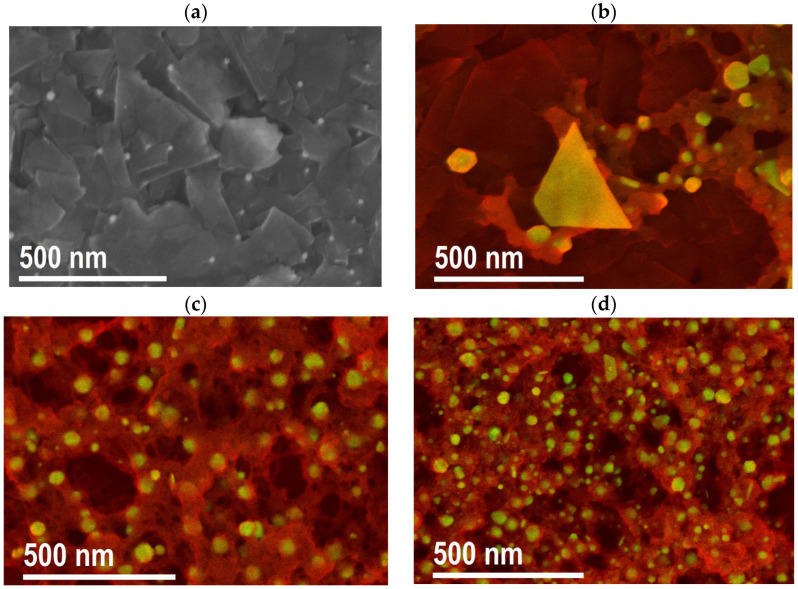
SEM micrographs of the AuNPs synthesized with different extracts of *Amphipterygium adstringens*, (**a**) Au@Mac-MeOH, (**b**) Au@US-EW, (**c**) Au@SWCD-EW, (**d**) Au@SWVD-EW.

**Figure 5 bioengineering-10-00437-f005:**
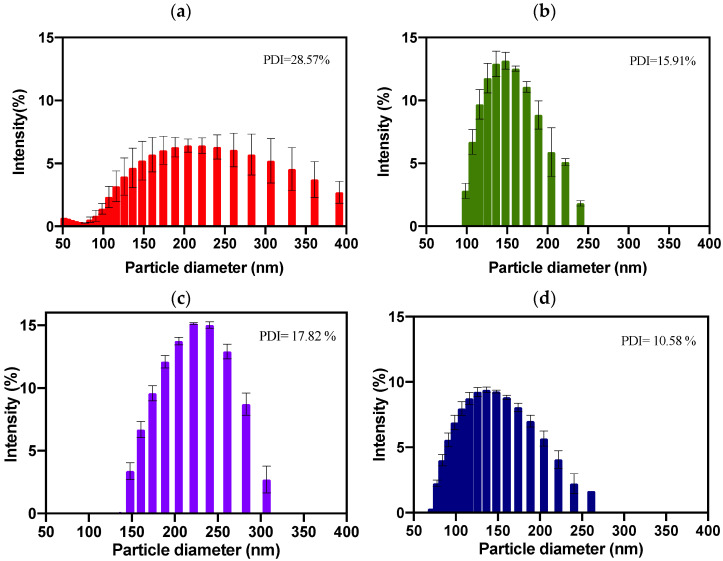
DLS histogram of AuNPs synthesized using different extracts from *Amphipterygium adstringens*. (**a**) Macerated with methanol (Mac-MeOH), (**b**) shock wave-assisted extraction with ethanol:water using constant delay (SWCD-EW), (**c**) shock wave-assisted extraction with ethanol:water using variable delay (SWVD-EW), (**d**) ultrasound-assisted extraction using water as a solvent (US-W).

**Figure 6 bioengineering-10-00437-f006:**
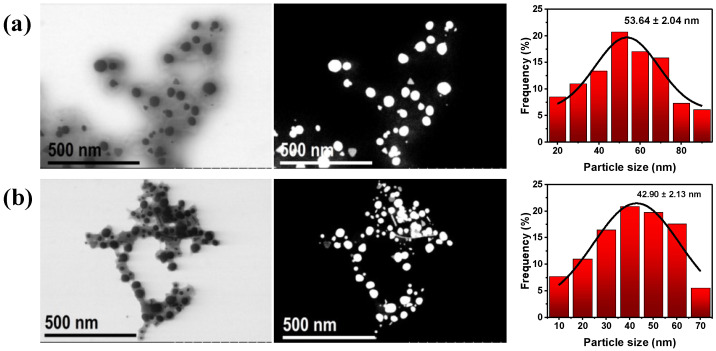
BF-STEM and ADF-STEM images and the corresponding histogram of (**a**) Au@SWCD-EW and (**b**) Au@US-W.

**Figure 7 bioengineering-10-00437-f007:**
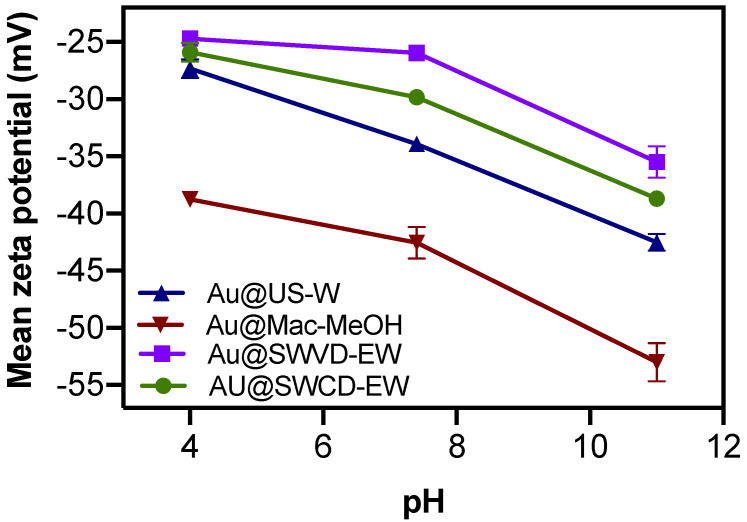
Z potential at different pH values (4, 7.4, and 11) of the AuNPs synthesized using different extracts of *Amphipterygium adstringens*.

**Figure 8 bioengineering-10-00437-f008:**
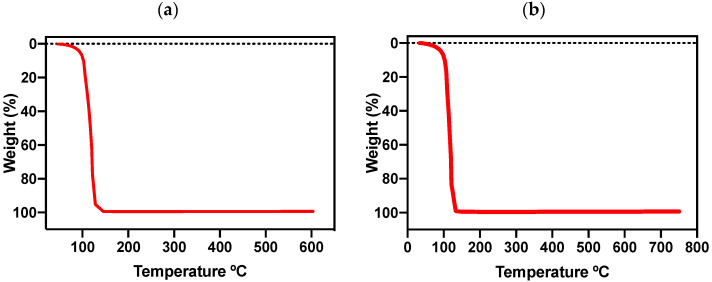
Thermogravimetric analysis of AuNPs synthesized with different *Amphipterygium adstringens* extracts (**a**) SWCD-EW, (**b**) US-W.

**Figure 9 bioengineering-10-00437-f009:**
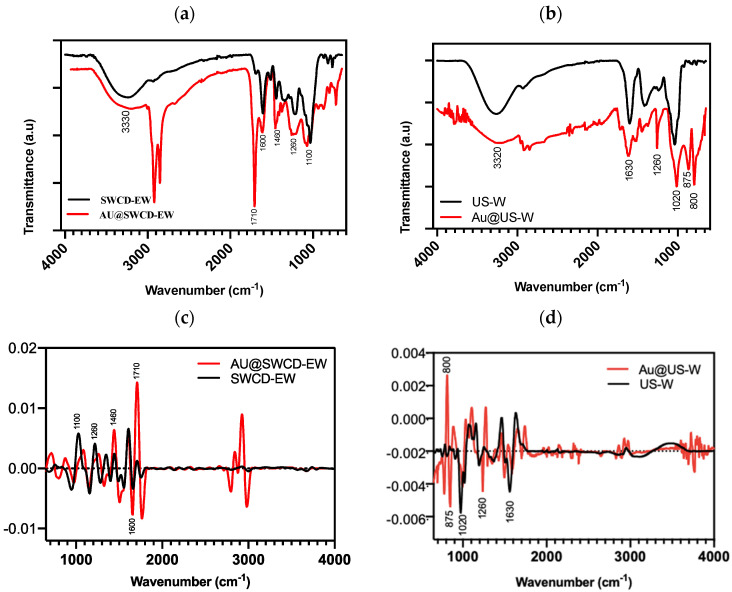
FTIR spectra of *A. adstringens* (**a**) shock wave extract and AuNPs (SWCD-EW and Au@SWCD-EW, respectively), and (**b**) ultrasound extract and AuNPs (US-W and Au@US-,W re-spectively). Second derivative of FTIR spectra of A. adstringens (**c**) shock wave extract and AuNPs (SWCD-EW and Au@SWCD-EW, respectively), and (**d**) ultrasound extract and AuNPS (US-W and Au@US-W, respectively).

**Figure 10 bioengineering-10-00437-f010:**
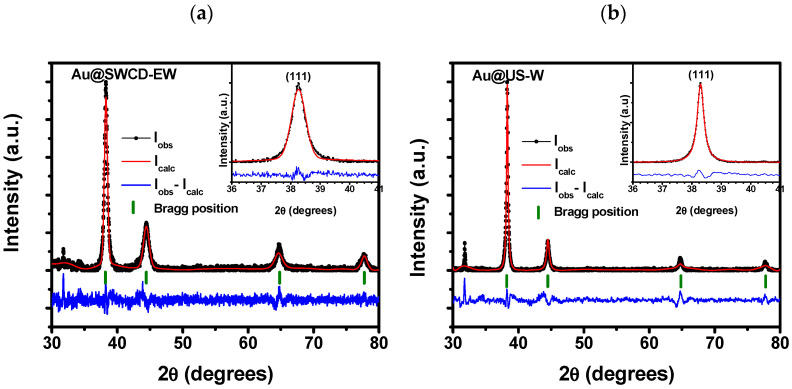
X-ray diffraction patterns of (**a**) Au@SWCD-EW and (**b**) Au@US-W.

**Figure 11 bioengineering-10-00437-f011:**
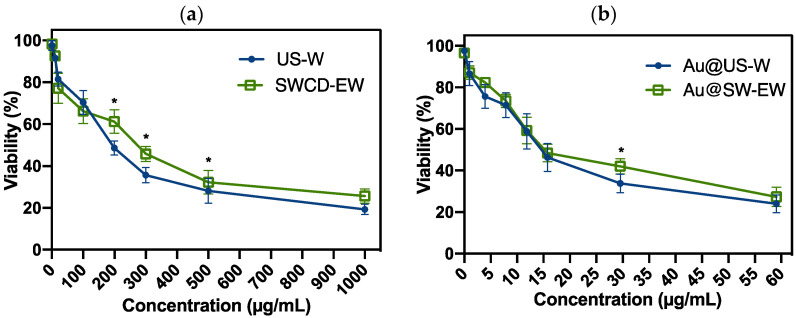
Dose-response curve of extracts and AuNPs at different doses determines the rIC_50_ in Jurkat cells. (**a**) US-W extract and SWCD-EW extract, (**b**) Au@US-W and Au@SWCD-EW. Mean +/− SE is shown at each point. The percentage of dead cells was identified by trypan blue assay. The statistically significant difference is indicated by *.

**Figure 12 bioengineering-10-00437-f012:**
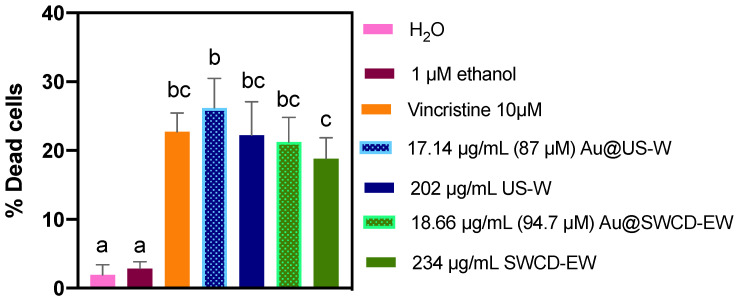
Cytotoxic effect of the negative control, positive control, and rIC_50_s of AuNPs and extracts tested in CRL-1991 cells. Mean +/− SE is shown in each column. According to one-way ANOVA and Tukey's multiple comparison tests, significant differences between groups are represented as different letters.

**Table 1 bioengineering-10-00437-t001:** Acronyms of the samples.

Sample	Acronym
*A. adstringens* methanolic macerate extraction	Mac-MeOH
*A. adstringens* ethanol:water ultrasound-assisted extraction	US-EW
*A. adstringens* ethanol:water shock wave-assisted extraction constant delay	SWCD-EW
*A. adstringens* ethanol:water shock wave-assisted extraction variable delay	SWVD-EW
*A. adstringens* water ultrasound-assisted extraction	US-W
AuNPs synthesized in *A. adstringens methanolic macerate extract*	Au@Mac-MeOH
AuNPs synthesized in *A. adstringens* ethanol:water ultrasound-assisted extraction	Au@US-EW
AuNPs synthesized in *A. adstringens* water ultrasound-assisted extraction	Au@US-W
AuNPs synthesized in *A. adstringens* ethanol:water shock wave-assisted extraction constant delay	Au@SWCD-EW
AuNPs synthesized in *A. adstringens* ethanol:water shock wave-assisted extraction variable delay	Au@SWVD-EW

**Table 2 bioengineering-10-00437-t002:** Total antioxidant capacity of samples extracted by different extraction methods.

Sample	TPC (mg/100 mLGAE) ^a^	TFC (mg/100 mL QE) ^b^	DPPH (mM/mL TEAC) ^c^	CUPRAC (mM/mL TEAC) ^d^
Mac-MeOH	30.27 ± 0.25	60.06 ± 0.14	1359.91 ± 23.38	2765 ± 11.05
US-EW	26.36 ± 0.43	75.41 ± 0.32	1836.72 ± 10.25	3018 ± 13.22
SWCD-EW	53.31 ± 0.37	68.58 ±0.25	4838.95 ± 39.23	7051 ± 18.24
SWVD-EW	39.15 ± 0.21	90.73 ± 0.18	2588.32 ± 50.17	5374 ± 36.44
US-W	57.83 ± 0.32	97.96 ± 0.41	3537.69 ± 34.81	5109 ± 27.92

^a^ Total phenolic content, ^b^ total flavonoid content, ^c^ 2,2-diphenyl-1-picrylhydrazyl, ^d^ cupric reducing antioxidant capacity.

## Data Availability

Data available on request due to privacy restrictions.

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
