# Peer review of "Green Synthesis and Antiproliferative Activity of Gold Nanoparticles of a Controlled Size and Shape Obtained Using Shock Wave Extracts from Amphipterygium adstringens"

_bioengineering, 2023, doi:10.3390/bioengineering10040437_

Round 1
Reviewer 1 Report
Green synthesis and antiproliferative activity of gold nanopar- 2 ticles of controlled size and shape obtained using shock wave 3 extracts from Amphipterygium adstringens
The authors performed the experiment with the concept of shock wave assisted extraction of A.adstringens.
The paper has the scientific quality for publication in the journal with the following recommendations
1. The effect of shock wave in plant extraction is need to be explain in detail (methodology, results)
2. Provide a graphical abstract for better understanding
3. Proofcheck the manuscript for grammatical errors and spell check
Author Response
Response to reviewer 1 comments
Point 1: The effect of shock wave in plant extraction is need to be explain in detail (methodology, results)
Point 1: This recommendation was followed; however, so far, there is still little information on the details of the phenomena involved in shock wave extraction. We believe that the requested explanation is not part of the methodology or our results. Because of this, it was included in the Introduction and as supplementary material (line 345-347).
- Provide a graphical abstract for better understanding:
Ponit 2: A graphical abstract has been added to the manuscript
- Proofcheck the manuscript for grammatical errors and spell check
Point 3: This requirement has been addressed.

Reviewer 2 Report
In the manuscript entitled “Green synthesis and antiproliferative activity of gold nanoparticles of controlled size and shape obtained using shock wave extracts from Amphipterygium adstringens”, Torres-Ortiz Daniela et al. have reported the use of green chemistry based on Amphipterygium adstringens as synthesis medium to obtain gold nanoparticles with controlled size and shape.
First of all, the abstract is too long; I suggest to remove the first paragraph and to start from “In this study, …”.
In the abstract, please, do not use undefined acronyms such as “Au@US-W” and “Au@SW-EW”; it is not clear to what have you referred to.
Please, add more references in the introduction.
In my opinion, details on the shock wave procedure and Figure 1 should be added in the experimental section.
Please, give the number to the equation reported in the experimental section, and add references.
For a better comprehension, I suggest overlapping the plots in a unique graphic, in Figure 4.
The legend of Figure 5 is poor comprehensive; please, define in a Table apart the used acronyms, at the beginning of the manuscript.
The English style is quite acceptable.
I can accept this manuscript with minor revisions.
Author Response
Response to reviewer 2 comments
Pont 1: First of all, the abstract is too long; I suggest to remove the first paragraph and to start from “In this study, …”.:
Response 1: Per suggestion, the abstract has been shortened
Point 2: In the abstract, please, do not use undefined acronyms such as “Au@US-W” and “Au@SW-EW”; it is not clear to what have you referred to:
Response 2: The acronyms have been removed from the abstract (line 12-24).
Point 3: Please, add more references in the introduction.
Response 3: More references have been added.
Point 4: In my opinion, details on the shock wave procedure and Figure 1 should be added in the experimental section.
Response 4: The observation is appreciated, though, authors decided that Figure 1, along with detailed shock wave information, should be relocated into supplementary material.
Point 5: Please, give the number to the equation reported in the experimental section, and add references.
Response 5: Your suggestion was followed.
Point 6: For a better comprehension, I suggest overlapping the plots in a unique graphic, in figure 4
Response 6: The plots have been overlapped.
Point 7: The legend of Figure 5 is poor comprehensive;
Response 7: Figure 5 has been simplified, as well as its legend.
Point 8: please, define in a Table apart the used acronyms, at the beginning of the manuscript. manuscript.
Response 8: The acronyms were defined in a separate table of the samples has been incorporated at the beginning of the manuscript.
Point 9: The English style is quite acceptable.
Point 10: I can accept this manuscript with minor revisions.

Reviewer 3 Report
Torres-Ortiz Daniela , García-Alcocer Guadalupe, Achim M. Loske , Fernández Francisco , Becerra-Becerra Edgardo , Esparza Rodrigo, Estevez Miriam:
Green synthesis and antiproliferative activity of gold nanoparticles of controlled size and shape obtained using shock wave extracts from Amphipterygium adstringens
The authors intend to present the biological effect of green synthesized gold nanoparticles on Jurkat cells using CRL-1991 cells as control system. The authors also present the methodical parts of nanoparticle production whereby the matter of the manuscript became very complex and not focused to the biological meaning. The authors used different methods but some application seems to be ad-hock measurements and do not provide solid data.
In details:
In Abstract: Symbols without meaning should not be used (Au@US-W, ..).
In Introduction the method of shock wave is presented in a long part. It belongs rather to the section of Method with its Figure 1 (or in supplementary). An extended table as a summary about the preparation steps, names/symbols of samples was strongly required. The chemical and compositional analysis of the extracts is fully missing.
In Results and Discussion
The description of the shock-wave extraction does not fit with the focus of the manuscript well. (The ultrasonic extraction has also important parameters. The authors can describe these technological details in supplementary.) The results about the chemical analysis of the different extract are strongly missing. The meaning of the symbols is hidden in the captions (remarks) of Table 1 and Figure 5. The samples are symbolized in different manner in the figures. The wavelength shifts in spectra due to the different sizes of Au-nanoparticles, is written (and well known), but the conclusion of the authors is missing. (WL, wavelength)
The SEM method provide an informative picturesque information, but with a very poor statistical reliability. (In Figure 5 no bars are given to show the size.) Consequently, I cannot accept the representation and interpretation of DLS measurements. (The intensity curves (in function of hydrodynamic radius ^6) or mass-distribution (in function of hydrodynamic radius ^3) are not given, the correlation function also not.) The presented number – particle diameter histograms seem to be truncated. The application of the thermogravimetric analysis is questionable for the determination of gold concentration. The spectra, shown in Fig. 9(a) are not overlap. A detailed study (using second derivatives) are needed to elucidate the coating from extracts in atomic (to show the perturbed vibration bands). (No wavelength in Fig.9, those are wavenumber.)
The reflection (111) is significantly sharper than that of other reflections. The interpretation is missing. (for the interpretation of the broadening of the reflection, a magnification, as an inset, is required) (presumably, instead of crystallite size, rather domain size).
Why were different concentration ranges used in cases a, b presented in Fig. 11? (how was converted the μg/mL to μM) What is the reproducibility of the measurements, especially the experiment with cells (applying freshly prepared coated/uncoated Au NPs) ?
Controlled size and shape mean that the authors elaborated a synthesis method whereby the desired parameters can be achieved. I am afraid, that these goals are not reached in the frame of the presented works. The manuscript cannot be accepted for publication in the present form. It should be rewritten in a more comprehensively form involving only the corrected and freshly obtained data.
Author Response
Response to reviewer 3 comments
Point 1: In Abstract: Symbols without meaning should not be used (Au@US-W, ..).
Response 1: All acronyms have been eliminated from the abstract.
Point 2: In Introduction the method of shock wave is presented in a long part. It belongs rather to the section of Method with its Figure 1 (or in supplementary).
Response 2: This observation is much appreciated, Figure 1 along with some shockwave detailed information, have been relocated to supplementary material.
Point 3: An extended table as a summary about the preparation steps, names/symbols of samples was strongly required.
Response 3: The acronyms were defined in a separate table of the samples has been incorporated at the beginning of the manuscript.
Point 4: The chemical and compositional analysis of the extracts is fully missing:
Response 4: For the approach of this study, only the antioxidant capacity of the extract is important to determine, as this parameter is fundamental for the green synthesis to take place. For this reason, antioxidant capacity by three different methods, total phenolic and flavonoid content are determined.
Point 5: The description of the shock-wave extraction does not fit with the focus of the manuscript well. (The ultrasonic extraction has also important parameters. The authors can describe these technological details in supplementary.)
Response 5: The manuscript has a subsection “3.1. Optimization of the parameters of the green extraction methods” that include ultrasonic parameters adjustment, nevertheless, this subsection has been relocated to materials and methods, now labeled 2.3. (line 346). Additionally, this information is incorporated to supplementary material.
Point 6: The results about the chemical analysis of the different extract are strongly missing:
Response 6: For the approach of this study, only the antioxidant capacity of the extract is important to determine, as this parameter is fundamental for the green synthesis to take place. For this reason, antioxidant capacity by three different methods, total phenolic and flavonoid content are determined.
Point 7: The meaning of the symbols is hidden in the captions (remarks) of Table 1 and Figure 5:
Response 7: As effectively suggested by the reviewer, a table with the meaning of the symbols has been added in the beginning of the manuscript.
Point 8: The samples are symbolized in different manner in the figures.
Response 8: This matter has been addressed.
Point 9: The wavelength shifts in spectra due to the different sizes of Au-nanoparticles, is written (and well known), but the conclusion of the authors is missing. (WL, wavelength).
Response 9: Discussion has been included. (line 697-700)
Point 10: The SEM method provide an informative picturesque information, but with a very poor statistical reliability. (In Figure 5 no bars are given to show the size.) Consequently, I cannot accept the representation and interpretation of DLS measurements. (The intensity curves (in function of hydrodynamic radius ^6) or mass-distribution (in function of hydrodynamic radius ^3) are not given, the correlation function also not.)
Response 10: Additional BF-STEM and DF-STEM experiments have been done in order to clear the difference of size between the hydrodynamic radios registered by the DLS assay, and the NPs size registered in the micrographs. Also, the conclusions have been adjusted.
Methodology: line 481-493
Results: 819-827
Point 11: The presented number – particle diameter histograms seem to be truncated.
Response 11: The range in the x axis of the histograms has been prolonged to display that histograms are not truncated, also the parameters of plotting were changed to Intensity in order to present clearer results (line 812).
Point 12: The application of the thermogravimetric analysis is questionable for the determination of gold concentration.
Response 12: References that support this information have been incorporated (line 865).
Point 13: The spectra, shown in Fig. 9(a) are not overlap. A detailed study (using second derivatives) are needed to elucidate the coating from extracts in atomic (to show the perturbed vibration bands). (No wavelength in Fig.9, those are wavenumber.)
Response 13: The IR spectra were not overlap, rather left with a slight separation between different IR spectra as it is more suitable this way to make a more accurate interpretation. Second derivative has been added.
Point 14: The reflection (111) is significantly sharper than that of other reflections. The interpretation is missing. (for the interpretation of the broadening of the reflection, a magnification, as an inset, is required) (presumably, instead of crystallite size, rather domain size).
Response 14: Rietveld refinement was computed to obtain more details about the structure, a new crystallite size was measurement, which is very simmilar to obtain by Scherrer equation.
The new figure shows inset the (111) reflection to observe the broadening of the reflections.
Point 15: Why were different concentration ranges used in cases a, b presented in Fig. 11? (how was converted the μg/mL to μM) What is the reproducibility of the measurements, especially the experiment with cells (applying freshly prepared coated/uncoated Au NPs) ?
Response 15: The range of concentration is different because in "a" we tested extracts in mg/mL while in "b" we tested de micromolar (mM) concentration of AuNPs after their characterization, and such concentration was determined as previously described in section 2.6 line 223 to 230. So, we did not convert mg/mL from the extracts to mM.
The reproducibility was determined by performing the viability assay by triplicate (each group) three times on different days. The SD was lower than 1.9 in each experiment. In each experiment, the AuNPs and extracts were freshly prepared.
Point 16: Controlled size and shape mean that the authors elaborated a synthesis method whereby the desired parameters can be achieved. I am afraid, that these goals are not reached in the frame of the presented works. The manuscript cannot be accepted for publication in the present form. It should be rewritten in a more comprehensively form involving only the corrected and freshly obtained data.

Reviewer 4 Report
Dear authors,
Greetings!
The manuscript “Green synthesis and antiproliferative activity of gold nanoparticles of controlled size and shape obtained using shock wave extracts from Amphipterygium adstringens” explores the synthesis of AuNPs using an eco-friendly protocol involving A. adstringens’ bark bought from a local market. Different solvents and techniques were applied to perform the extraction. The nanomaterials produced were characterized and the ones that met pre-stablished patterns were assayed to verify cytotoxicity.
Adjustments and improvements are necessary in order to improve scientific soundness and quality.
Regarding “Abstract”, the words “we” and “our” can be removed to avoid using the first person. The sentence “Gold nanoparticles (AuNPs) with sizes ranging between 100 and 150 nm were obtained with ultrasound aqueous” seems to lack some words at the end; it needs attention.
When it comes to “Introduction”, the second paragraph lacks references as same as the second sentence from paragraph three. In line 112, “this study” is used twice; the sentence needs improvement. The content presented from the end of line 119 to line 123 should be removed once it refers to conclusion and not introduction.
Regarding “Materials and Methods” to validate the study it is necessary to prove that the material bought from “a local market” is A. adstringens. Who identified the material as A. adstringens? Any analysis of phytochemicals was performed to verify the identity of the material? In line 139, I believe the intention was to write “Germany” instead of “Alemania”. Please mention the supplier of alcohols used. The methanolic extract is not mentioned neither in item “2.2” nor in abstract; it is necessary to fix that. In line 175, meq should be written as mEq. Lines from 254 to 261 would fit better in a new section dedicated to AuNPs’ cytotoxicity assay, to be presented before the item dedicated to “trypan blue assay”. It is necessary to add a subsection dedicated to statistics.
When it comes to “Results” section, text presented from line 265 to 275 does not refer to results; it refers to “Materials and Methods” section and needs to be placed in correct section. In subsection “3.2” appears for the first time a not mentioned before methanolic extract. It is necessary to adapt the whole manuscript to explain this extract correctly. It is necessary to explain the meaning of Mac-MeOH too before presenting the abbreviation. Regarding the UV-Vis analysis, according to subsection 2.6 it was performed in samples of the extract after synthesis. So, it is necessary to present 3 results from UV-Vis analysis: extract with NPs, supernatant (extract only) after centrifugation and resuspended NPs (after centrifugation). Only by doing that it will be possible to verify if the band attributed to plasmon resonance is from NPs or a band from metabolites present in the extract. Regarding size difference when comparing DLS results and SEM results: it was not mentioned steps to clean the synthesized NPs; it is necessary to wash NPs and centrifuge until a clean supernatant is obtained. Then they can be resuspended in type I water and sonicated before preparation for microscopy. Please, provide TEM images of NPs in a zoom capable of allowing secure measurement based on the scale bar. These procedures will probably guarantee a better alignment of DLS and TEM results. The NPs not being clean enough can affect not only the image’s obtainment but also thermogravimetric analysis, as observed. Section 3.6 seems out of place and presents no results; it should be moved to “Materials and Methods” and the observations that guided grouping should be presented and explained. Please, show the results from line 269 and discuss them.
Regarding “Conclusions”: It was not proved that Au@SWCD-EW were size-controlled and measured 50 nm. SEM demonstrated a medium size between 50 and 100 nm and DLS demonstrated an average size of 110nm. It is necessary to adjust experiments and conclusion. It is also important to add future perspectives at the end of it.
Figure 3 needs improvements; it is hard to see the scale bar inside it. Please provide the bar outside the round area and over a black background to become easier to see. Figure 5 also needs improvements to allow better visualization of the scale bar; and in 5.a3 a letter “A” is missing before “u@SWCD-M”. Figure 11 lacks control and statistics.
English language and style are fine/minor spell check required.
Author Response
Response to reviewer 4 comments
Point 1: Regarding “Abstract”, the words “we” and “our” can be removed to avoid using the first person. The sentence “Gold nanoparticles (AuNPs) with sizes ranging between 100 and 150 nm were obtained with ultrasound aqueous” seems to lack some words at the end; it needs attention.
Response 1: Observations have been attended (lines 12-24)
Point 2: When it comes to “Introduction”, the second paragraph lacks references as same as the second sentence from paragraph three.
Response 2: References have been added.
Point 3: In line 112, “this study” is used twice; the sentence needs improvement.
Response 3 This duplicate has been eliminated.
Point 4: The content presented from the end of line 119 to line 123 should be removed once it refers to conclusion and not introduction.
Response 4: The suggestion has been taken into consideration.
Point 5: Regarding “Materials and Methods” to validate the study it is necessary to prove that the material bought from “a local market” is A. adstringens. Who identified the material as A. adstringens? Any analysis of phytochemicals was performed to verify the identity of the material?
Response 5: No phytochemical analysis is required as plant specimen are biological identified by a biologist taking into consideration morphological characteristics, comparison with the stored specimen in the herbal, anatomic studies, and literature review (line 237-239).
Point 6: In line 139, I believe the intention was to write “Germany” instead of “Alemania”. Please mention the supplier of alcohols used.
Response 6: This issue has been attended (line 246).
Point 7: The methanolic extract is not mentioned neither in item “2.2” nor in abstract; it is necessary to fix that.
Response 7: Mention of the methanolic extract has been introduced to the manuscript in the abstract, and materials and methods (line 17-18, 315).
Point 8: In line 175, meq should be written as mEq.
Response 8: This issue has been attended (line 418).
Point 9: Lines from 254 to 261 would fit better in a new section dedicated to AuNPs’ cytotoxicity assay, to be presented before the item dedicated to “trypan blue assay”. It is necessary to add a subsection dedicated to statistics.
Response 9: Instead of adding a new section, we changed the name of subsection 2.7.2 for “Cytotoxicity assay with Trypan blue” to fit better lines 254 to 261. Subsection 2.7.3 was added and dedicated to statistics (line 502 and 572-577)
Point 10: When it comes to “Results” section, text presented from line 265 to 275 does not refer to results; it refers to “Materials and Methods” section and needs to be placed in correct section.
Response 10: The whole subsection “3.1. Optimization of the parameters of the green extraction methods” has been relocated to materials and methods, now labeled 2.3 (line 343).
Point 11: In subsection “3.2” appears for the first time a not mentioned before methanolic extract. It is necessary to adapt the whole manuscript to explain this extract correctly. It is necessary to explain the meaning of Mac-MeOH too before presenting the abbreviation.
Response 11: Methanolic extract has been incorporated to the abstract and materials and methods, also an acronym table has been included at the beginning of the manuscript.
Point 12: Regarding the UV-Vis analysis, according to subsection 2.6 it was performed in samples of the extract after synthesis. So, it is necessary to present 3 results from UV-Vis analysis: extract with NPs, supernatant (extract only) after centrifugation and resuspended NPs (after centrifugation). Only by doing that it will be possible to verify if the band attributed to plasmon resonance is from NPs or a band from metabolites present in the extract.
Response 12: UV-Vis spectra of the extract previous to the synthesis has been included, alongside AuNPs spectra (line 697).
Point 13: Regarding size difference when comparing DLS results and SEM results: it was not mentioned steps to clean the synthesized NPs; it is necessary to wash NPs and centrifuge until a clean supernatant is obtained. Then they can be resuspended in type I water and sonicated before preparation for microscopy.
Response 13: Cleaning process is described in section 2.6, nevertheless, the explanation has been modified to an improved version (line 271-273).
Point 14: Please, provide TEM images of NPs in a zoom capable of allowing secure measurement based on the scale bar. These procedures will probably guarantee a better alignment of DLS and TEM results. The NPs not being clean enough can affect not only the image’s obtainment but also thermogravimetric analysis, as observed.
Response 14: A scale bar has been included to each SEM micrographs, and new STEM images of thoroughly cleaned AuNPs, BF-STEM and DF-STEM experiments have been done in order to clear the difference of size between the hydrodynamic radios registered by the DLS assay, and the NPs size registered in the micrographs. Also, the conclusions have been adjusted (line 818-827).
Point 15: Section 3.6 seems out of place and presents no results; it should be moved to “Materials and Methods” and the observations that guided grouping should be presented and explained.
Response 15: Authors appreciated this observation and decided to wipe out this section, as it doesn´t seem relevant.
Point 16: Please, show the results from line 269 and discuss them.
Response 16: These results have been incorporated to the supplementary material section.
Point 17: Regarding “Conclusions”: It was not proved that Au@SWCD-EW were size-controlled and measured 50 nm. SEM demonstrated a medium size between 50 and 100 nm and DLS demonstrated an average size of 110nm. It is necessary to adjust experiments and conclusion.
Response 17: Additional BF-STEM and DF-STEM experiments have been done in order to clear the difference of size between the hydrodynamic radios registered by the DLS assay, and the NPs size registered in the micrographs. Also, the conclusions have been adjusted.
Point 18: It is also important to add future perspectives at the end of it.
Response 18: Perspectives have been included (line 1123-1127).
Point 19: Figure 3 needs improvements; it is hard to see the scale bar inside it. Please provide the bar outside the round area and over a black background to become easier to see.
Response 19: This issue has been taken care of, the bar has been substituted by a more visible line (Figure 3 has been changed to Figure 2).
Point 20: Figure 5 also needs improvements to allow better visualization of the scale bar; and in 5.a3 a letter “A” is missing before “u@SWCD-M”.
Response 20: Currently it is Figure 4, a scale bar has been added, and only 4 micrographs samples are presented, the rest have been moved to supplementary material.
Point 21: Figure 11 lacks control and statistics.
Response 21: Figure 11 includes negative controls which are the concentration 0 mg/mL and 0 mM from the extracts and AuNPs, respectively. In each point of the dose-response and concentration-response curves the means +/- SEs are shown.
English language and style are fine/minor spell check required.

Round 2
Reviewer 3 Report
Torres-Ortiz Daniela , García-Alcocer Guadalupe, Achim M. Loske , Fernández Francisco , Becerra-Becerra Edgardo , Esparza Rodrigo, Estevez Miriam:
Green synthesis and antiproliferative activity of gold nanoparticles of controlled size and shape obtained using shock wave extracts from Amphipterygium adstringens
The manuscript is improved significantly after the review. It, however, contains details to correct or change.
In Fig 3 (a), (b) the four sample cannot be distinguished (2 red and 2 blue curves appear).
“The size of the AuNP samples agree with the results obtained from the antioxidant capacity; because the US-W extract had the higher antioxidant capacity of the four samples, it can reduce more Au+3 to Au0, resulting in larger NPs.” It may be true or not. (The question is still open: enormous number of small NPs can also be occurred. The effect of the colloidal formations (self- assembly) are also important in the precursor extraction.)
The DLS intensities are plotted, what is the reason for this representation? (The volume, mass of NPs is more convenient.) The BF-TEM and DF-TEM images are very informative (please do not use “STEM”, because it refers to the apparat/dual electron-microscope).
The thermogravimetric analysis is not a sophisticated method, as it was mentioned in the first review. To search the valence state/s of Au is recommended by the means of XPS. The thermal analysis, executed in high temperature range, can induce the formation of carbonaceous material which are thermal stable. (How was the measurement carried out in air or inert gas?)
The reflection (111) is significantly sharper than that of other reflections. Presumably, the NPs have more domains with different extension. (With other words, a single particle is not a single crystal, it contains more domains.) The broadening is due to the reflection of small crystallites inside particles. Consequently, the broadening may not inform us about the particle size (especially the domain size is significantly smaller than the particle size. (I suggest the application of small angle X-ray scattering in the future.)
One question is still open, how was converted the μg/mL to μM?
Please insert the Table with samples and acronyms in the section Materials and Methods
Author Response
Point 1: In Fig 3 (a), (b) the four sample cannot be distinguished (2 red and 2 blue curves appear).
Response 1: In figure 3, the colors in the curves have been reassigned.
Point 2: “The size of the AuNP samples agree with the results obtained from the antioxidant capacity; because the US-W extract had the higher antioxidant capacity of the four samples, it can reduce more Au+3 to Au0, resulting in larger NPs.” It may be true or not. (The question is still open: enormous number of small NPs can also be occurred. The effect of the colloidal formations (self- assembly) are also important in the precursor extraction.)
Response 2: The authors agree with this comment.
Point 3: The DLS intensities are plotted, what is the reason for this representation? (The volume, mass of NPs is more convenient.)
Response 3: The size distribution obtained from a measurement is based on intensity for all DLS equipment. Nevertheless, using the Mie theory is possible to convert the intensity distribution into the volume or number distribution when the particles are considerably smaller than the wavelength (in our case, lower tjan 300nm) of the illuminating light and when the absorption and refractive index of the particles are known. This step was performed because the obtained average size of the particles by microscopy is always closest to the number distribution from light scattering. However, because the intensity distributions highlight the larger particles in the distribution while the number distributions highlight the smaller ones, the distribution by intensity showed a single smooth peak, while the number distribution provided a clear asymmetric peak that produced to another reviewer some doubts.
Point 4: The BF-TEM and DF-TEM images are very informative (please do not use “STEM”, because it refers to the apparat/dual electron-microscope).
Response 4: The images were acquiring in a SEM/STEM dual microscope, therefore use “STEM” is correct, BF-TEM is almost similar to BF-STEM, however, DF-TEM is different to DF-STEM, we change DF-STEM by ADF-STEM that is the correct technique to indicate the Z-contrast.
Point 5: The thermogravimetric analysis is not a sophisticated method, as it was mentioned in the first review. To search the valence state/s of Au is recommended by the means of XPS. The thermal analysis, executed in high temperature range, can induce the formation of carbonaceous material which are thermal stable. (How was the measurement carried out in air or inert gas?)
Response 5: The analysis was made in air atmosphere. The authors agree with the comment, we had a mis redaction, the data obtained by the TGA correspond to the AuNPs, that are most likely structured by Au0 and Au+1, as have been reported in various studies. XPS analysis of the AuNPs has been conducted and added to the manuscript and to the supplementary material.
Point 6: The reflection (111) is significantly sharper than that of other reflections. Presumably, the NPs have more domains with different extension. (With other words, a single particle is not a single crystal, it contains more domains.) The broadening is due to the reflection of small crystallites inside particles. Consequently, the broadening may not inform us about the particle size (especially the domain size is significantly smaller than the particle size. (I suggest the application of small angle X-ray scattering in the future.)
Response 6: the referee is right, the presence of facetted particles (domains) in the sample explains the discrepancy of the broadening among the different reflections. We delete the part where compare crystallite size (XRD) with particle size (SEM). SAXS is good technique to analyse dispersion or nanoparticles colloidal, however we don’t have the equipment, but we will try to perform this technique in a future.
Point 7: One question is still open, how was converted the μg/mL to μM?
Response 7: The extracts were not converted to µM. The µM expression corresponds to the metallic concentration present in the AuNPs per mL. To present a homogenize style regarding extract and AuNPs concentration, the µg/mL relation has been added to the molar concentration.
Point 8: Please insert the Table with samples and acronyms in the section Materials and Methods
Response 8: The table has been repositioned to materials and methods.
Reviewer 4 Report
Dear authors,
greetings!
The manuscript was largely improved in quality. The only aspect still missing, is to present in Figure 11 (using asterisks or letters as in Figure 12, for example) statistics.
Author Response
Point 1: The manuscript was largely improved in quality. The only aspect still missing, is to present in Figure 11 (using asterisks or letters as in Figure 12, for example) statistics.
Response 1: Asterisks have been added to Figure 11.